# Distributed Principal Component Analysis with Limited Communication

**Foivos Alimisis**
Department of Mathematics
University of Geneva

**Peter Davies**
Department of Computer Science
University of Surrey

**Bart Vandereycken**
Department of Mathematics
University of Geneva

**Dan Alistarh**
IST Austria &
Neural Magic, Inc.

## Abstract

We study efficient distributed algorithms for the fundamental problem of principal component analysis and leading eigenvector computation on the sphere, when the data are randomly distributed among a set of computational nodes. We propose a new quantized variant of Riemannian gradient descent to solve this problem, and prove that the algorithm converges with high probability under a set of necessary spherical-convexity properties. We give bounds on the number of bits transmitted by the algorithm under common initialization schemes, and investigate the dependency on the problem dimension in each case.

## 1 Introduction

The need to process ever increasing quantities of data has motivated significant research interest into efficient *distributed* algorithms for standard tasks in optimization and data analysis; see, e.g., [16, 27, 11]. In this context, one of the fundamental metrics of interest is *communication cost*, measured as the number of bits sent and received by the processors throughout computation. Communication-efficient variants are known for several classical tasks and algorithms, from basic ones such as mean estimation [28, 20, 8] to variants of gradient descent [19, 12] or even higher-order optimization [2, 15].

By contrast, much less is known concerning the communication cost for classical problems in data analysis, such as Principal Component Analysis (PCA) [13, 5], or spectral clustering [23]. In this paper, we will focus on the closely-related problem of computing the leading eigenvector of the data covariance matrix, i.e., determining the first principal component, in the setting where $d$-dimensional data comes from an unknown distribution, and is split among $n$ machines.

Given the fundamental nature of PCA, there is a rich body of work on solving *sequential* variants, both in the Euclidean domain [24, 25, 26] and, more recently, based on Riemannian optimization. This is because maximizing the Rayleigh quotient can be re-interpreted in terms of a hyper-sphere, leading to an unconstrained Riemannian optimization problem [1, 32, 31]. These references have exclusively focused on the *sequential* setting, where the data can be stored and processed by a single node. However, in many settings the large volume of data requires processing by multiple nodes, which has inspired significant work on reducing the distribution overheads of PCA, e.g. [4, 11], specifically on obtaining efficient algorithms in terms of *round complexity*.

More precisely, an impressive line of work considered the *deterministic* setting [17, 4, 6, 10], where the input is partitioned arbitrarily between machines, for which both round-efficient and communication-efficient solutions are now known. The general approach is to first perform a careful low-rank approximation of the local data and then communicate iteratively to reconstruct a faithful

estimate of the covariance matrix. Due to the worst-case setting, the number of communication rounds scales polynomially in the eigengap $\delta$ and in the precision $\varepsilon$, making them costly in practice.

Thus, on the practical side, it is common to assume a setting with stochastic rather than worst-case data partitioning. Specifically, Garber et al. [11] consider this setting, and note that standard solutions based on distributed Power Method or Lanczos would require a number of communication rounds which increases with the sample size, which is impractical at scale. They circumvent this issue by replacing explicit Power Method iterations with a set of convex optimization problems, which can be solved in a round-efficient manner. Huang and Pan [14] proposed a round-efficient solution, by leveraging the connection to Riemannian optimization. Their proposal relies on a procedure which allows the update of node-local variables via a surrogate gradient, whose deviation from the global gradient should be bounded. Thus, communication could only be used to reach consensus on the local variables among the nodes. Unfortunately, upon close inspection, we have noted significant technical gaps in their analysis, which we discuss in detail in the next section.

The line of work in the stochastic setting focused primarily on *latency cost*, i.e., the number of communication rounds for solving this problem. For highly-dimensional data and/or highly-distributed settings, it is known that *bandwidth cost*, i.e., the number of bits which need to be transmitted per round, can be a significant bottleneck, and an important amount of work deals with it for other classic problems, e.g. [28, 12, 15]. By contrast, much less is known concerning the communication complexity of distributed PCA.

**Contribution.** In this paper, we provide a new algorithm for distributed leading eigenvector computation, which specifically minimizes the total number of bits sent and received by the nodes. Our approach leverages the connection between PCA and Riemannian optimization; specifically, we propose the first distributed variant of Riemannian gradient descent, which supports *quantized communication*. We prove convergence of this algorithm utilizing a set of necessary convexity-type properties, and bound its communication complexity under different initializations.

At the technical level, our result is based on a new *sequential* variant of Riemannian gradient descent on the sphere. This may be of general interest, and the algorithm can be generalized to Riemannian manifolds of bounded sectional curvatures using more sophisticated geometric bounds. We strengthen the gradient dominance property proved in [31], by proving that our objective is weakly-quasi-convex in a ball of some minimizer and combining that with a known quadratic-growth condition. We leverage these properties by adapting results of [7] in the Riemannian setting and choosing carefully the learning-rate in order to guarantee convergence for the single-node version.

This algorithm is specifically-designed to be amenable to distribution: specifically, we port it to the distributed setting, and add communication-compression in the tangent spaces of the sphere, using a variant of the lattice quantization technique of [8], which we port from the context of standard gradient descent. We provide a non-trivial analysis for the communication complexity of the resulting algorithm under different initializations. We show that, under random initialization, a solution can be reached with total communication that is quadratic in the dimension $d$, and linear in $1/\delta$, where $\delta$ is the gap between the two biggest eigenvalues; when the initialization is done to the leading eigenvector of one of the local covariance matrices, this dependency can be reduced to linear in $d$.

## 2   Setting and Related Work

**Setting.** We assume to be given $m$ total samples coming from some distribution $\mathcal{D}$, organized as a global $m \times d$ data matrix $M$, which is partitioned row-wise among $n$ processors, with node $i$ being assigned the matrix $M_i$, consisting of $m_i$ consecutive rows, such that $\sum_{i=1}^{n} m_i = m$. As is common, let $A := M^T M = \sum_{i=1}^{d} M_i^T M_i$ be the global covariance matrix, and $A_i := M_i^T M_i$ the local covariance matrix owned by node $i$. We denote by $\lambda_1, \lambda_2, ..., \lambda_d$ the eigenvalues of $A$ in descending order and by $v_1, v_2, ..., v_d$ the corresponding eigenvectors. Then, it is well-known that, if the gap $\delta = \lambda_1 - \lambda_2$ between the two leading eigenvalues of $A$ is positive, we can approximate the leading eigenvector by solving the following empirical risk minimization problem up to accuracy $\varepsilon$:

$$x^\star = \mathrm{argmin}_{x \in \mathbb{R}^d} \left( -\frac{x^T A x}{\|x\|^2} \right) = \mathrm{argmin}_{x \in \mathbb{R}^d, \|x\|_2 = 1} \left( -x^T A x \right) = \mathrm{argmin}_{x \in \mathbb{S}^{d-1}} \left( -x^T A x \right), \tag{1}$$

where $\mathbb{S}^{d-1}$ is the $(d-1)$-dimensional sphere.

We define $f : \mathbb{S}^{d-1} \to \mathbb{R}$, with $f(x) = -x^T A x$ and $f_i : \mathbb{S}^{d-1} \to \mathbb{R}$, with $f_i(x) = -x^T A_i x$. Since the inner product is bilinear, we can write the global cost as the sum of the local costs: $f(x) = \sum_{i=1}^{n} f_i(x)$.

In our analysis we use notions from the geometry of the sphere, which we recall in Appendix A.

**Related Work.** As discussed, there has been a significant amount of research on efficient variants of PCA and related problems [5, 24, 25, 26, 1, 32, 31]. Due to space constraints, we focus on related work on communication-efficient algorithms. In particular, we discuss the relationship to previous *round-efficient* algorithms; to our knowledge, this is the first work to specifically focus on the bit complexity of this problem in the setting where data is randomly partitioned. More precisely, previous work on this variant implicitly assumes that algorithms are able to transmit real numbers at unit cost.

The straightforward approach to solve the minimization problem (1) in a distributed setting, where the data rows are partitioned, would be to use a distributed version of the Power Method, Riemannian gradient descent (RGD), or the Lanczos algorithm. In order to achieve an $\varepsilon$-approximation of the minimizer $x^\star$, the latter two algorithms require $\tilde{\mathcal{O}}(\lambda_1 \log(1/\varepsilon)/\delta)$ rounds, where the $\tilde{\mathcal{O}}$ notation hides poly-logarithmic factors in $d$. Distributed Lanczos and accelerated RGD would improve this by an $\mathcal{O}(\sqrt{\lambda_1/\delta})$ factor. However, Garber et al. [11] point out that, when $\delta$ is small, e.g. $\delta = \Theta(1/\sqrt{m})$, then unfortunately the number of communication rounds would increase with the sample size, which renders these algorithms non-scalable.

Standard distributed convex approaches, e.g. [16, 27], do not directly extend to our setting due to non-convexity and the unit-norm constraint. Garber et al. [11] proposed a variant of the Power Method called *Distributed Shift-and-Invert (DSI)* which converges in roughly $\mathcal{O}\left(\sqrt{\frac{b}{\delta\sqrt{m}}} \log^2(1/\varepsilon) \log(1/\delta)\right)$ rounds, where $b$ is a bound on the squared $\ell_2$-norm of the data. Huang and Pan [14] aimed to improve the dependency of the algorithm on $\varepsilon$ and $\delta$, and proposed an algorithm called *Communication-Efficient Distributed Riemannian Eigensolver (CEDRE)*. This algorithm is shown to have round complexity $\mathcal{O}(\frac{b}{\delta\sqrt{m}} \log(1/\varepsilon))$, which does not scale with the sample size for $\delta = \Omega(1/\sqrt{m})$, and has only logarithmic dependency on the accuracy $\varepsilon$.

**Technical issues in [14].** Huang and Pan [14] proposed an interesting approach, which could provide the most round-efficient distributed algorithm to date. Despite the fact that we find the main idea of this paper very creative, we have unfortunately identified a significant gap in their analysis, which we now outline.

Specifically, one of their main results, Theorem 3, uses the local gradient dominance shown in [31]; yet, the application of this result is invalid, as it is done without knowing in advance that the iterates of the algorithms continue to remain in the ball of initialization. This is compounded by another error on the constant of gradient dominance (Lemma 2), which we believe is caused by a typo in the last part of the proof of Theorem 4 in [31]. This typo is unfortunately propagated into their proof. More precisely, the objective is indeed gradient dominated, but with a constant which vanishes when we approach the equator, in contrast with the $2/\delta$ which is claimed *globally*. (We reprove this result independently in Proposition 4). Thus, starting from a point lying in some ball of the minimizer can lead to a new point where the objective is more poorly gradient dominated.

This is a non-trivial technical problem which, in the case of gradient descent, can be addressed by a choice of the learning rate depending on the initialization. Given these issues, we perform a new and formally-correct analysis for gradient descent based on new convexity properties which guarantee convergence directly in terms of iterates, and not just function values. We would like however to note that our focus in this paper is on bit and not round complexity.

# 3 Step 1: Computing the Leading Eigenvector in One Node

## 3.1 Convexity-like Properties and Smoothness

We first analyze the case that all the matrices $A_i$ are simultaneously known to all machines. This practically means that we can run gradient descent in one of the machines (let it be the master), since we have all the data stored there. All the proofs for this section are in Appendix B.

Our problem reads

$$\min_{x \in \mathbb{S}^{d-1}} -x^T A x$$

where $A = M^T M$ is the global covariance matrix. If $\delta = \lambda_1 - \lambda_2 > 0$, this problem has exactly two global minima: $v_1$ and $-v_1$. Let $x \in \mathbb{S}^{d-1}$ be an arbitrary point. Then $x$ can be written in the form $x = \sum_{i=1}^{d} \alpha_i v_i$. Fixing the minimizer $v_1$, we have that a ball in $\mathbb{S}^{d-1}$ around $v_1$ is of the form

$$B_a = \{x \in \mathbb{S}^{d-1} \mid \alpha_1 \geq a\} = \{x \in \mathbb{S}^{d-1} \mid \langle x, v_1 \rangle \geq a\} \tag{2}$$

for some $a$. Without loss of generality, we may assume $a > 0$ (otherwise, consider the ball around $-v_1$ to establish convergence to $-v_1$). In this case, the region $\mathcal{A}$ as defined in Def. 11 is identical to $B_a$ and $x^* := v_1$.

We begin by investigating the convexity properties of the function $-x^T A x$. In particular, we prove that this function is weakly-quasi-convex (see Def. 13) with constant $2a$ in the ball $B_a$. See also Appendix A for a definition of the Riemannian gradient $\mathrm{grad} f$ and the logarithm $\log$.

**Proposition 1.** *The function $f(x) = -x^T A x$ satisfies*

$$2a(f(x) - f^*) \leq \langle \mathrm{grad} f(x), -\log_x(x^*) \rangle$$

*for any $x \in B_a$ with $a > 0$.*

We continue by providing a quadratic growth condition for our cost function, which can also be found in [30, Lemma 2] in a slightly different form. Here dist is the intrinsic distance in the sphere, that we also define in Appendix A.

**Proposition 2.** *The function $f(x) = -x^T A x$ satisfies*

$$f(x) - f^* \geq \frac{\delta}{4} \mathrm{dist}^2(x, x^*)$$

*for any $x \in B_a$ with $a > 0$.*

According to Def. 14, this means that $f$ satisfies quadratic growth with constant $\mu = \delta/2$.

Next, we prove that quadratic growth and weak-quasi-convexity imply a "strong-weak-convexity" property. This is similar to [7, Proposition 3.1].

**Proposition 3.** *A function $f \colon \mathbb{S}^{d-1} \to \mathbb{R}$ which satisfies quadratic growth with constant $\mu > 0$ and is $2a$-weakly-quasi-convex with respect to a minimizer $x^*$ in a ball of this minimizer, satisfies for any $x$ in the same ball the inequality*

$$f(x) - f^* \leq \frac{1}{a} \langle \mathrm{grad} f(x), -\log_x(x^*) \rangle - \frac{\mu}{2} \mathrm{dist}^2(x, x^*).$$

Interestingly, using Proposition 3, we can recover the gradient dominance property proved in [31, Theorem 4].

**Proposition 4.** *The function $f(x) = -x^T A x$ satisfies*

$$\|\mathrm{grad} f(x)\|^2 \geq \delta a^2 (f(x) - f^*)$$

*for any $x \in B_a$ with $a > 0$.*

The proof of Proposition 4 can be done independently based only on Proposition 3 and [7, Lemma 3.2]. This verifies that our results until now are optimal regarding the quality of initialization since gradient dominance proved in [31] is optimal in that sense (i.e. the inequalities in the proof are tight).

**Proposition 5.** *The function $f(x) = -x^T A x$ is geodesically $2\lambda_1$-smooth on the sphere.*

Thus, the smoothness constant of $f$, as defined in Def. 12, equals $\gamma = 2\lambda_1$. Similarly, let $\gamma_i$ denote the smoothness constant of $f_i$, which equals twice the largest eigenvalue of $A_i$. In order to estimate $\gamma$ in a distributed fashion using only the local data matrices $A_i$, we shall use the over-approximation $\gamma \leq n \max_{i=1,\ldots,n} \gamma_i$.

## 3.2 Convergence

We now consider Riemannian gradient descent with learning rate $\eta > 0$ starting from a point $x^{(0)} \in B_a$:

$$x^{(t+1)} = \exp_{x^{(t)}}(-\eta \operatorname{grad} f(x^{(t)})),$$

where $\exp$ is the exponential map of the sphere, defined in Appendix A.

Using Proposition 3 and a proper choice of $\eta$, we can establish a convergence rate for the instrinsic distance of the iterates to the minimizer.

**Proposition 6.** *An iterate of Riemannian gradient descent for $f(x) = -x^T A x$ starting from a point $x^{(t)} \in B_a$ and with step-size $\eta \leq a/\gamma$ where $\gamma \geq 2\lambda_1$, produces a point $x^{(t+1)}$ that satisfies*

$$\operatorname{dist}^2(x^{(t+1)}, x^*) \leq (1 - a\mu\eta)\operatorname{dist}^2(x^{(t)}, x^*) \qquad \text{for } \mu = \delta/2.$$

Note that this result implies directly that if our initialization $x^{(0)}$ lies in the ball $B_a$, the distance of $x^{(1)}$ to the center $x^*$ decreases and thus all subsequent iterates continue being in the initialization ball. This is essential since it guarantees that the convexity-like properties for $f$ continue to hold during the whole optimization process. The proof of Proposition 6 is in Appendix B.

This result is also valid in a more general setting when applied to an arbitrary $f \colon \mathbb{S}^{d-1} \to \mathbb{R}$ that is $2a$-weakly-quasi convex and $\gamma$-smooth, and that satisfies quadratic growth with constant $\mu$. The Euclidean analogue of this result can be found in [7, Lemma 4.4].

## 4 Step 2: Computing the Leading Eigenvector in Several Nodes

We now present our version of distributed gradient descent for leading eigenvector computation and measure its bit complexity until reaching accuracy $\epsilon$ in terms of the intrinsic distance of an iterate $x^{(T)}$ from the minimizer $x^*$ ($x^*$ is the leading eigenvector closest to the initialization point).

**Lattice Quantization.** For estimating the Riemannian gradient in a distributed manner with limited communication, we use a quantization procedure developed in [8]. The original quantization scheme involves randomness, but we use a *deterministic* version of it, by picking up the closest point to the vector that we want to encode. This is similar to the quantization scheme used by [3] and has the following properties.

**Proposition 7.** *[8, 3] Denoting by $b$ the number of bits that each machine uses to communicate, there exists a quantization function*

$$Q \colon \mathbb{R}^{d-1} \times \mathbb{R}^{d-1} \times \mathbb{R}_+ \times \mathbb{R}_+ \to \mathbb{R}^{d-1},$$

*which, for each $w, y > 0$, consists of an encoding function $\operatorname{enc}_{w,y} \colon \mathbb{R}^{d-1} \to \{0,1\}^b$ and a decoding one $\operatorname{dec}_{w,y} \colon \{0,1\}^b \times \mathbb{R}^{d-1} \to \mathbb{R}^{d-1}$, such that, for all $x, x' \in \mathbb{R}^{d-1}$,*

- *$\operatorname{dec}_{w,y}(\operatorname{enc}_{w,y}(x), x') = Q(x, x', y, w)$, if $\|x - x'\| \leq y$.*

- *$\|Q(x, x', y, w) - x\| \leq w$, if $\|x - x'\| \leq y$.*

- *If $y/w > 1$, the cost of the quantization procedure in number of bits satisfies $b = \mathcal{O}((d-1)\log_2\left(\frac{y}{w}\right)) = \mathcal{O}(d\log\left(\frac{y}{w}\right))$.*

In the following, the quantization takes place in the tangent space of each iterate $T_{x^{(t)}}\mathbb{S}^{d-1}$, which is linearly isomorphic to $\mathbb{R}^{d-1}$. We denote by $Q_x$ the specification of the function $Q$ at $T_x\mathbb{S}^{d-1}$. The vector inputs of the function $Q_x$ are represented in the local coordinate system of the tangent space that the quantization takes place at each step. For decoding at $t > 0$, we use information obtained in the previous step, that we need to translate to the same tangent space. We do that using parallel transport $\Gamma$, which we define in Appendix A.

**Algorithm**

We present now our main algorithm, which is inspired by quantized gradient descent firstly designed by [22], and its similar version in [3].

1. Choose an arbitrary machine to be the master node, let it be $i_0$.

2. Choose $x^{(0)} \in \mathbb{S}^{d-1}$ (we analyze later specific ways to do that).

3. Consider the following parameters

$$\sigma := 1 - \cos(D)\mu\eta, \ K := \frac{2}{\sqrt{\sigma}}, \ \theta := \frac{\sqrt{\sigma}(1 - \sqrt{\sigma})}{4},$$

$$\sqrt{\xi} := \theta K + \sqrt{\sigma}, \ R^{(t)} = \gamma K(\sqrt{\xi})^t D$$

where $D$ is an over-approximation for $\mathrm{dist}(x^{(0)}, x^*)$ (the way to practically choose $D$ is discussed later).

We assume that $\cos(D)\mu\eta \leq \frac{1}{2}$, otherwise we run the algorithm with $\sigma = \frac{1}{2}$.

In $T_{x^{(0)}}\mathbb{S}^{d-1}$:

4. Compute the local Riemannian gradient $\mathrm{grad} f_i(x^{(0)})$ at $x^{(0)}$ in each node.

5. Encode $\mathrm{grad} f_i(x^{(0)})$ in each node and decode in the master node using its local information:

$$q_i^{(0)} = Q_{x^{(0)}}\left(\mathrm{grad} f_i(x^{(0)}), \mathrm{grad} f_{i_0}(x^{(0)}), 2\gamma\pi, \frac{\theta R^{(0)}}{2n}\right).$$

6. Sum the decoded vectors in the master node:

$$r^{(0)} = \sum_{i=1}^{n} \mathrm{grad} f_i(x^{(0)}).$$

7. Encode the sum in the master and decode in each machine $i$ using its local information:

$$q^{(0)} = Q_{x^{(0)}}\left(r^{(0)}, \mathrm{grad} f_i(x^{(0)}), \frac{\theta R^{(0)}}{2} + 2\gamma\pi, \frac{\theta R^{(0)}}{2}\right).$$

For $t \geq 0$:

8. Take a gradient step using the exponential map:

$$x^{(t+1)} = \exp_{x^{(t)}}(-\eta q^{(t)})$$

with step-size $\eta$ (the step-size is discussed later).

In $T_{x^{(t+1)}}\mathbb{S}^{d-1}$:

9. Compute the local Riemannian gradient $\mathrm{grad} f_i(x^{(t+1)})$ at $x^{(t+1)}$ in each node.

10. Encode $\mathrm{grad} f_i(x^{(t+1)})$ in each node and decode in the master node using its (parallelly transported) local information from the previous step:

$$q_i^{(t+1)} = Q_{x^{(t+1)}}\left(\mathrm{grad} f_i(x^{(t+1)}), \Gamma_{x^{(t)}}^{x^{(t+1)}} q_i^{(t)}, \frac{R^{(t+1)}}{n}, \frac{\theta R^{(t+1)}}{2n}\right).$$

11. Sum the decoded vectors in the master node:

$$r^{(t+1)} = \sum_{i=1}^{n} \mathrm{grad} f_i(x^{(t+1)}).$$

12. Encode the sum in the master and decode in each machine using its local information in the previous step after parallel transport:

$$q^{(t+1)} = Q_{x^{(t+1)}}\left(r^{(t+1)}, \Gamma_{x^{(t)}}^{x^{(t+1)}} q^{(t)}, \left(1 + \frac{\theta}{2}\right)R^{(t+1)}, \frac{\theta R^{(t+1)}}{2}\right).$$

---

**Convergence** We first control the convergence of iterates simultaneously with the convergence of quantized gradients.

Note that

$$\sqrt{\xi} = \frac{1 - \sqrt{\sigma}}{2} + \sqrt{\sigma} = \frac{1 + \sqrt{\sigma}}{2} \leq \frac{\sqrt{2}\sqrt{1 + \sigma}}{2} = \sqrt{\frac{1 + \sigma}{2}}.$$

**Lemma 8.** *If $\eta \leq \frac{\cos(D)}{\gamma}$, the previous quantized gradient descent algorithm produces iterates $x^{(t)}$ and quantized gradients $q^{(t)}$ that satisfy*

$$\text{dist}^2(x^{(t)}, x^*) \leq \xi^t D^2, \quad \|q_i^{(t)} - \text{grad} f_i(x^{(t)})\| \leq \frac{\theta R^{(t)}}{2n}, \quad \|q^{(t)} - \text{grad} f(x^{(t)})\| \leq \theta R^{(t)}.$$

The proof is a Riemannian adaptation of the similar one in [22] and [3] and is presented in Appendix C. We recall that since the sphere is positively curved, it provides a landscape easier for optimization. It is quite direct to derive a general Riemannian method for manifolds of bounded curvature using more advanced geometric bounds, however this exceeds the scope of this work which focuses on leading eigenvector computation.

We now move to our main complexity result.

**Theorem 9.** *Let $\eta \leq \frac{\cos(D)}{\gamma}$. Then, the previous quantized gradient descent algorithm needs at most*

$$b = \mathcal{O}\left(nd\frac{1}{\cos(D)\delta\eta} \log\left(\frac{n}{\cos(D)\delta\eta}\right) \log\left(\frac{D}{\epsilon}\right)\right)$$

*bits in total to estimate the leading eigenvector with an accuracy $\epsilon$ measured in intrinsic distance.*

The proof is based on the previous Lemma 8 in order to count the number of steps that the algorithm needs to estimate the minimizer with accuracy $\epsilon$ and Proposition 7 to count the quantization cost in each round. We present the proof again in Appendix C.

## 5   Step 3: Dependence on Initialization

### 5.1   Uniformly random initialization

The cheapest choice to initialize quantized gradient descent is a point in the sphere chosen uniformly at random. According to Theorem 4 in [31], such a random point $x^{(0)}$ will lie, with probability at least $1 - p$, in a ball $B_a$ (see Def. 2) where

$$a \geq c\frac{p}{\sqrt{d}} \iff \text{dist}(x^{(0)}, x^*) \leq \arccos\left(c\frac{p}{\sqrt{d}}\right). \tag{3}$$

Here, $c$ is a universal constant (we estimated numerically that $c$ is lower bounded by 1, see Appendix D, thus we can use $c = 1$). By choosing the step-size $\eta$ as

$$\eta = \frac{cp}{\sqrt{d}\gamma}$$

and the parameter $D$ as

$$D = \arccos\left(c\frac{p}{\sqrt{d}}\right),$$

we are guaranteed that $\eta = \frac{\cos(D)}{\gamma}$, and $\text{dist}(x^{(0)}, x^*) \leq D$ with probability at least $1 - p$. Our analysis above therefore applies (up to probability $1 - p$) and the general communication complexity result becomes

$$\mathcal{O}\left(\frac{nd}{\cos(D)\delta\eta} \log \frac{n}{\cos(D)\delta\eta} \log \frac{D}{\epsilon}\right) = \mathcal{O}\left(\frac{nd}{\eta\gamma\delta\eta} \log \frac{n}{\eta\gamma\delta\eta} \log \frac{D}{\epsilon}\right) = \mathcal{O}\left(\frac{nd}{\eta^2\gamma\delta} \log \frac{n}{\eta^2\gamma\delta} \log \frac{D}{\epsilon}\right).$$

Substituting $\eta^2 = \frac{p^2 c^2}{d\gamma^2}$, the number of bits satisfies (up to probability $1 - p$) the upper bound

$$b = \mathcal{O}\left(n\frac{d^2}{p^2}\frac{\gamma}{\delta} \log \frac{nd\gamma}{p\delta} \log \frac{D}{\epsilon}\right) = \tilde{\mathcal{O}}\left(n\frac{d^2}{p^2}\frac{\gamma}{\delta}\right).$$

## 5.2 Warm Start

A reasonable strategy to get a more accurate initialization is to perform an eigenvalue decomposition to one of the local covariance matrices, for instance $A_{i_0}$ (in the master node $i_0$), and compute its leading eigenvector, let it be $v^{i_0}$. For simplicity we will assume here that each machine hosts the same number of data points $m_i = \frac{m}{n}$. Then we communicate $v^{i_0}$ to all machines in order to use the normalized quantized approximation $x^{(0)}$ as initialization. We define:

$$\tilde{x}^{(0)} = Q\left(v^{i_0}, v^i, \|v^{i_0} - v^i\|, \frac{\langle v^{i_0}, x^* \rangle}{2(\sqrt{2} + 2)}\right)$$

$$x^{(0)} = \frac{\tilde{x}^{(0)}}{\|\tilde{x}^{(0)}\|}$$

where $Q$ is the lattice quantization scheme in $\mathbb{R}^d$ (i.e. we quantize the leading eigenvector of the master node as a vector in $\mathbb{R}^d$ and then project back to the sphere). The input and output variance in this quantization can be bounded by constants that we can practically estimate (see Appendix D).

**Proposition 10.** *Assume that our data are i.i.d. and sampled from a distribution $\mathcal{D}$ bounded in $\ell_2$ norm by a constant $h$. Given that the eigengap $\delta$, the number of machines $n$ and the total number of data points $m$ satisfy*

$$\delta \geq \Omega\left(\sqrt{m}\sqrt{n}\sqrt{\log \frac{d}{p}}\right), \tag{4}$$

*we have that the previous quantization costs $\mathcal{O}(nd)$ many bits and $\langle x^{(0)}, x^* \rangle$ is lower bounded by a constant with probability at least $1 - p$.*

Thus, if bound 4 is satisfied, then the communication complexity becomes

$$b = \mathcal{O}\left(nd\frac{\gamma}{\delta}\log\frac{n\gamma}{\delta}\log\frac{1}{\epsilon}\right) = \tilde{\mathcal{O}}\left(nd\frac{\gamma}{\delta}\right)$$

many bits in total with probability at least $1 - p$ (notice that this can be further simplified using bound 4). This is because $D$ in Theorem 9 is upper bounded by a constant and the communication cost of quantizing $v^{i_0}$ does not affect the total communication cost. If we can estimate the specific relation in bound 4 (the constant hidden inside $\Omega$), then we can compute estimations of the quantization parameters in the definition of $x^{(0)}$.

Condition 4 is quite typical in this literature; see [14] and references therein (beginning of page 2), as we also briefly discussed in the introduction. Notice that $\sqrt{m}$ appears in the numerator and not the denominator, only because we deal with the sum of local covariance matrices and not the average, thus our eigengap is $m$ times larger than the eigengap of the normalized covariance matrix. Denoting by $\delta'$ the eigengap of the normalized covariance matrix, bound 4 can be written equivalently as

$$\delta' \geq \Omega\left(\frac{\sqrt{n}}{\sqrt{m}}\sqrt{\log\frac{d}{p}}\right) = \tilde{\Omega}\left(\frac{1}{\sqrt{m_{i_0}}}\right),$$

where $m_{i_0}$ is the number of data points owned by the master node (and any other machine) and $\tilde{\Omega}$ hides logarithmic factors from the lower bound.

## 6 Experiments

We evaluate our approach experimentally, comparing the proposed method of Riemannian gradient quantization against three other benchmark methods:

- Full-precision Riemannian gradient descent: Riemannian gradient descent, as described in Section 3.2, is performed with the vectors communicated at full (64-bit) precision.

- Euclidean gradient difference quantization: the 'naïve' approach to quantizing Riemannian gradient descent. Euclidean gradients are quantized and averaged before being projected to Riemannian gradients and used to take a step. To improve performance, rather than quantizing Euclidean gradients directly, we quantize the difference between the current local gradient and the previous local gradient, at each node. Since these differences are generally smaller than the gradients themselves, we expect this quantization to introduce lower error.

- Quantized power iteration: we also use as a benchmark a quantized version of power iteration, a common method for leading-eigenvector computation given by the update rule $x^{(t+1)} \leftarrow \frac{Ax^{(t)}}{\|Ax^{(t)}\|}$. $Ax^{(t)}$ can be computed in distributed fashion by communicating and summing the vectors $A_i x^{(t)}, i \leq n$. It is these vectors that we quantize.

All three of the quantized methods use the same vector quantization routine, for fair comparison.

**Figure 1** Convergence results on real datasets

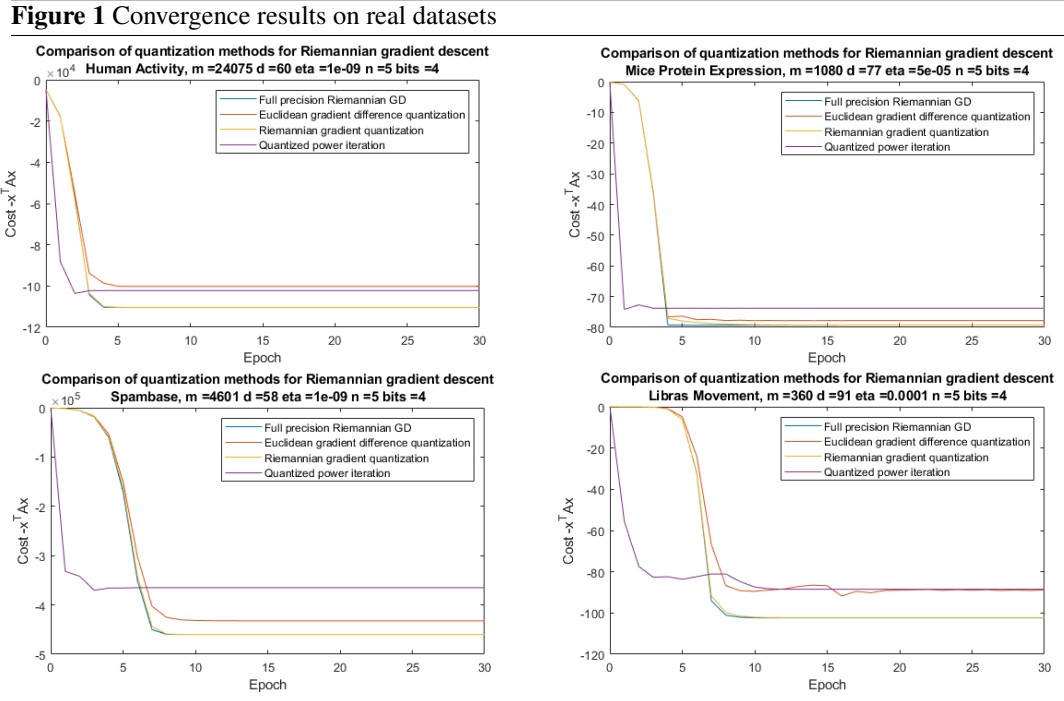

We show convergence results (Figure 1) for the methods on four real datasets: Human Activity from the MATLAB Statistics and Machine Learning Toolbox, and Mice Protein Expression, Spambase, and Libras Movement from the UCI Machine Learning Repository [9]. All results are averages over 10 runs of the cost function $-x^T Ax$ (for each method, the iterates $x$ are normalized to lie in the 1-ball, so a lower value of $-x^T Ax$ corresponds to being closer to the principal eigenvector).

All four datasets display similar behavior: our approach of Riemannian gradient quantization outperforms naïve Euclidean gradient quantization, and essentially matches the performance of the full-precision method while communicating only 4 bits per coordinate, for a $16\times$ compression. Power iteration converges slightly faster (as would be expected from the theoretical convergence guarantees), but is much more adversely affected by quantization, and reaches a significantly suboptimal result.

We also tested other compression levels, though we do not present figures here due to space constraints: for fewer bits per coordinate (1-3), the performance differences between the four methods are in the same order but more pronounced; however, their convergence curves become more erratic due to increased variance from quantization. For increased number of bits per coordinate, the performance of the quantized methods reaches that of the full-precision methods at between 8-10 bits per coordinate depending on input, and the observed effect of quantized power iteration converging to a suboptimal result gradually diminishes. Our code is publicly available [1].

---

[1] https://github.com/IST-DASLab/QRGD

# 7 Discussion and Future Work

We proposed a communication efficient algorithm to compute the leading eigenvector of a covariance data matrix in a distributed fashion using only limited amount of bits. To the best of our knowledge, this is the first algorithm for this problem with bounded bit complexity. We focused on gradient descent since it provides robust convergence in terms of iterates, and can circumvent issues that arise due to a cost function that only has local convexity-like properties; second, it known to be amenable to quantization.

We would like briefly to discuss the round complexity of our method, emphasizing inherent round complexity trade-offs due to quantization. Indeed, the round complexity of our quantized RGD algorithm for accuracy $\epsilon$ is $\tilde{\mathcal{O}}(\frac{\gamma}{a^2\delta})$, where $\epsilon$ appears inside a logarithm. Specifically, if the initialization $x^{(0)}$ is uniformly random in the sphere (section 5.1), then $a = \mathcal{O}(1/\sqrt{d})$ . If we have warm start (section 5.2), then $a = \mathcal{O}(1)$. Both statements hold with high probability.

For random initialization, the round complexity of distributed power method is $\tilde{\mathcal{O}}(\frac{\lambda_1}{\delta}) = \tilde{\mathcal{O}}(\frac{\gamma}{\delta})$ (see the table at the beginning of page 4 in [11]). This is because the convergence rate of power method is $\tan(\text{dist}(x_k, x^*)) \leq (1 - \lambda_1/\delta)^k \tan(\text{dist}(x_0, x^*))$. The dimension is again present in that rate, because $\tan(\text{dist}(x_0, x^*)) = \Theta(\sqrt{d})$, but it appears inside a logarithm in the final iteration complexity.

Since we want to insert quantization, the standard way to do so is by starting from a rate which is contracting (see e.g. [22]). It may in theory be possible to add quantization in a non-contracting algorithm, but we are unaware of such a technique. Thus, we firstly prove that RGD with step-size $\frac{a}{\gamma}$ is contracting (based on weak-convexity and quadratic growth) and then we add quantization, paying the price of having an extra $\mathcal{O}(1/a^2) = \mathcal{O}(d)$ in our round complexity. Thus, our round complexity is slightly worse than distributed power method, but this is in some sense inherent due to the introduction of quantization. Despite this, our algorithm can achieve low bit complexity and it is more robust in perturbations caused by bit limitations, as our experiments suggest.

We conclude with some interesting points for future work.

First, our choice of step-size is quite conservative since the local convexity properties of the cost function improve when approaching the optimum. Thus, instead of setting the step-size to be $\eta = a/\gamma$, where $a$ is a lower bound for $\langle x^{(0)}, x^* \rangle$, we can re-run the whole analysis with an adaptive step-size $\eta = \langle x^{(t)}, x^* \rangle / \gamma$ based on an improved value for $a$ from $\langle x^{(t)}, x^* \rangle$. This should improve the dependency on the dimension of our complexity results.

Second, the version of Riemannian gradient descent that we have chosen is expensive, since one needs to calculate the exponential map in each step using sin and cos calculations. Practitioners usually use cheaper retraction-based versions, and it would be of interest to study how our analysis can be modified if the exponential map is substituted by a retraction (i.e., any first-order approximation).

Third, it would be interesting to see whether the ideas and techniques presented in this work can also be used for other data science problems in a distributed fashion through communication-efficient Riemannian methods. This could be, for example, PCA for the top $k$ eigenvectors using optimization on the Stiefel or Grassmann manifold [1] and low-rank matrix completion on the fixed-rank manifold [29].

## Acknowledgments and Disclosure of Funding

We would like to thank the anonymous reviewers for helpful comments and suggestions. We also thank Aurelien Lucchi and Antonio Orvieto for fruitful discussions at an early stage of this work. FA is partially supported by the SNSF under research project No. 192363 and conducted part of this work while at IST Austria under the European Union's Horizon 2020 research and innovation programme (grant agreement No. 805223 ScaleML). PD partly conducted this work while at IST Austria and was supported by the European Union's Horizon 2020 programme under the Marie Skłodowska-Curie grant agreement No. 754411.

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
