# A    Geometry of the Sphere

We analyze here briefly some basic notions of the geometry of the sphere that we use in our algorithm and convergence analysis. We refer the reader to [1, p. 73–76] for a more comprehensive presentation.

**Tangent Space:** The tangent space of the $r$-dimensional sphere $\mathbb{S}^r$ at a point $p$ is an $r$-dimensional vector space, which generalizes the notion of tangent plane in two dimensions. We denote it by $T_p\mathbb{S}^r$ and a vector $v$ belongs in it, if and only if, it can be written as $\dot{\alpha}(0)$, where $\alpha\colon (-\varepsilon, \varepsilon) \to \mathbb{S}^r$ (for some $\varepsilon > 0$) is a smooth curve with $\alpha(0) = p$. The tangent space at $p$ can be given also in an explicit way, as the set of all vectors in $\mathbb{R}^{r+1}$ orthogonal to $p$ with respect to the usual inner product. Given a vector $w \in \mathbb{R}^{r+1}$, we can always project it orthogonally in any tangent space of $\mathbb{S}^r$. Taking all vectors to be column vectors, the orthogonal projection in $T_p\mathbb{S}^r$ satisfies

$$P_p(w) = (I - pp^T)w.$$

**Geodesics:** Geodesics on high-dimensional surfaces are defined to be locally length-minimizing curves. On the $d$-dimensional sphere, they coincide with great circles. These can be computed explicitly and give rise to the exponential and logarithmic maps. The **exponential map** at a point $p$ is defined as $\exp_p : T_p\mathbb{S}^r \to \mathbb{S}^r$ with $\exp_p(v) = c(1)$, with $c$ being the (unique) geodesic (i.e., **length minimizing part** of great circle) satisfying $c(0) = p$ and $\dot{c}(0) = v$. Defining the exponential map in this way makes it invertible with inverse $\log_p$. The exponential and logarithmic map are given by well-known explicit formulas:

$$\exp_p(v) = \cos(\|v\|)p + \sin(\|v\|)\frac{v}{\|v\|}, \quad \log_p(q) = \arccos(\langle q, p\rangle)\frac{P_p(q-p)}{\|P_p(q-p)\|}. \tag{5}$$

The distance between points $p$ and $q$ measured intrinsically on the sphere is

$$\mathrm{dist}(p, q) = \|\log_p(q)\| = \arccos(\langle q, p\rangle). \tag{6}$$

Notice that $\langle q, p\rangle = \|q\|\|p\| \cos(\angle(p, q)) = \cos(\angle(p, q))$, thus the distance of $p$ and $q$ is actually the angle between them.

The inner product inherited by the ambient Euclidean space $\mathbb{R}^{r+1}$ provides a way to transport vectors from a tangent space to another one, parallelly to the geodesics. This operation is called parallel transport and constitutes an orthogonal isomorphism between the two tangent spaces. We denote $\Gamma_p^q$ for the parallel transport from $T_pM$ to $T_qM$ along the geodesic connecting $p$ and $q$. If $q = \exp_p(tv)$, then parallel transport is given by the formula

$$\Gamma_p^q u = \left(I_d + \cos(t\|v\|)-1)\frac{vv^T}{\|v\|^2} - \sin(t\|v\|)\frac{pv^t}{\|v\|}\right)u.$$

**Riemannian Gradient:** Given a function $f\colon \mathbb{S}^r \to \mathbb{R}$, we can define its differential at a point $p$ in a tangent direction $v \in T_p\mathbb{S}^r$ as

$$df(p)v = \lim_{t\to 0}\frac{f(\alpha(t)) - f(p)}{t},$$

where $\alpha\colon (-\varepsilon, \varepsilon) \to \mathbb{S}^r$ is a smooth curve defined such that $\alpha(0) = p$ and $\dot{\alpha}(0) = v$. Using this intrinsically defined Riemannian differential, we can define the Riemannian gradient at $p$ as a vector $\mathrm{grad} f(p) \in T_p\mathbb{S}^r$, such that

$$\langle \mathrm{grad} f(p), v\rangle = df(p)v,$$

for any $v \in T_p\mathbb{S}^r$. In the case of the sphere, we can also compute the Riemannian gradient by orthogonally projecting the Euclidean gradient $\nabla f(p)$ computed in the ambient space into the tangent space of $p$:

$$\mathrm{grad} f(p) = P_p(\nabla f(p)).$$

**Geodesic Convexity and Smoothness:** In this paragraph we define convexity-like properties in the context of the sphere, which we employ in our analysis.

**Definition 11.** *A subset $\mathcal{A} \subseteq \mathbb{S}^r$ is called geodesically uniquely convex, if every two points in $\mathcal{A}$ are connected by a unique geodesic.*

Open hemispheres satisfy this definition, thus they are geodesically uniquely convex. Actually, they are the biggest possible subsets of the sphere with this property.

**Definition 12.** *A smooth function $f \colon \mathcal{A} \to \mathbb{R}$ is called geodesically $\gamma$-smooth if*

$$\|\mathrm{grad} f(p) - \Gamma_q^p \mathrm{grad} f(q)\| \le \gamma \, \mathrm{dist}(p, q)$$

*for any $p, q \in \mathcal{A}$.*

**Definition 13.** *A smooth function $f \colon \mathcal{A} \to \mathbb{R}$ defined in a geodesically uniquely convex subset $\mathcal{A} \subset \mathbb{S}^r$ is called geodesically $a$-weakly-quasi-convex (for some $a > 0$) with respect to a point $p \in A$ if*

$$a(f(x) - f(p)) \le \langle \mathrm{grad} f(x), -\log_x(p) \rangle \qquad \text{for all } x \in \mathcal{A}.$$

Observe that weak-quasi-convexity implies that any local minimum of $f$ lying in the interior of $\mathcal{A}$ is also a global one.

**Definition 14.** *A function $f \colon \mathcal{A} \to \mathbb{R}$ is said to satisfy quadratic growth condition with constant $\mu$, if*

$$f(x) - f^* \ge \frac{\mu}{2} \mathrm{dist}^2(x, x^*) \qquad \text{for all } x \in \mathcal{A},$$

*where $f^*$ is the minimum of $f$ and $x^*$ the global minimizer of $f$ closest to $x$.*

Quadratic growth condition is slightly weaker than the so-called gradient dominance one:

**Definition 15.** *A function $f \colon \mathcal{A} \to \mathbb{R}$ is said to be $\mu$-gradient dominated if*

$$f(x) - f^* \le \frac{1}{2\mu} \|\mathrm{grad} f(x)\|^2 \qquad \text{for all } x \in \mathcal{A}.$$

Gradient dominance with constant $\mu$ implies quadratic growth with the same constant. The proof can be done by a direct Riemannian adaptation of the argument from [18, p. 13–14].

**Tangent Space Distortion:** The only non-trivial geometric result we use in this work is that the geodesics of the sphere spread more slowly than in a flat space. This is a direct application of the Rauch comparison theorem since the sphere has constant positive curvature.

**Proposition 16.** *Let $\Delta abc$ a geodesic triangle on the sphere. Then we have*

$$\mathrm{dist}(a, b) \le \|\log_c(a) - \log_c(b)\|.$$

Notice that in Euclidean space we have equality in this bound.

## B  Properties of the Cost and Convergence of Single-Node Gradient Descent

In this appendix, we will prove the convergence of Riemannian gradient descent applied to

$$\min_{x \in \mathbb{S}^{d-1}} f(x) \qquad \text{with } f(x) = -x^T A x$$

and $A \in \mathbb{R}^{d \times d}$ a symmetric and positive definite matrix. In the following, we will denote the eigenvalues of $A$ by $\lambda_1 > \lambda_2 \ge \cdots \ge \lambda_d > 0$ and their corresponding orthonormal eigenvectors by $v_1, v_2, \ldots, v_d$. Denote the spectral gap by $\delta = \lambda_1 - \lambda_2 > 0$.

Let $f^* = -\lambda_1$ denote the minimum of $f$ on $\mathbb{S}^{d-1}$ which is attained at $x^* = v_1$.

**Proposition 1.** *The function $f(x) = -x^T A x$ satisfies*

$$2a(f(x) - f^*) \le \langle \mathrm{grad} f(x), -\log_x(x^*) \rangle$$

*for any $x \in B_a$ with $a > 0$.*

*Proof.* For any $x \in B_a$, we can write

$$x = \sum_{i=1}^d \alpha_i v_i, \qquad Ax = \sum_{i=1}^d \lambda_i \alpha_i v_i \tag{7}$$

for some scalars $\alpha_i$. Recall that $\alpha_1 \geq a > 0$ by definition (2) of $B_a$.

With the orthogonal projector $P_x = I - xx^T$ onto the tangent space $T_x\mathbb{S}^{d-1}$, we get from (5) that

$$\langle \text{grad} f(x), -\log_x(x^*)\rangle = \langle P_x \nabla f(x), \frac{\text{dist}(x, x^*)}{\|P_x(x - x^*)\|} P_x(x - x^*)\rangle$$

$$= \frac{\text{dist}(x, x^*)}{\|P_x(x - x^*)\|} \langle P_x \nabla f(x), x - x^*\rangle.$$

because $P_x^2 = P_x$.

Direct calculation now gives

$$\langle P_x \nabla f(x), x - x^*\rangle = -2x^T Ax + 2\langle Ax, x^*\rangle - 2f(x)\|x\|^2 + 2f(x)\langle x, x^*\rangle$$

$$= 2f(x) + 2\lambda_1\alpha_1 - 2f(x) + 2f(x)\alpha_1$$

$$= 2\alpha_1(f(x) + \lambda_1) = 2\alpha_1(f(x) - f^*) \geq 0.$$

It is easy to verify that $\text{dist}(x, x^*) \geq \|P_x(x - x^*)\|$. We thus obtain

$$\langle \text{grad} f(x), -\log_x(x^*)\rangle \geq 2\alpha_1(f(x) - f^*),$$

which gives the desired result since $\alpha_1 \geq a$. $\qquad\square$

**Proposition 2.** *The function $f(x) = -x^T Ax$ satisfies*

$$f(x) - f^* \geq \frac{\delta}{4}\text{dist}^2(x, x^*)$$

*for any $x \in B_a$ with $a > 0$.*

*Proof.* The proof follows the one in [30, Lemma 2]. Using the expansions in (7), we get

$$x^T Ax = \sum_{i=1}^d \lambda_i \alpha_i^2 = \lambda_1 \alpha_1^2 + \sum_{i=2}^d \lambda_i \alpha_i^2 \leq \lambda_1 \alpha_1^2 + \lambda_2(1 - \alpha_1^2)$$

since $\|x\|^2 = 1 = \sum_{i=1}^d \alpha_i^2$. From (6), we have that $\alpha_1 = \cos(\text{dist}(x, x^*))$ and so

$$x^T Ax \leq \lambda_1 \cos^2(\text{dist}(x, x^*)) + \lambda_2 \sin^2(\text{dist}(x, x^*)).$$

Direct calculation now shows

$$f(x) - f^* = -x^T Ax + \lambda_1 \geq \lambda_1 - \lambda_1 \cos^2(\text{dist}(x, x^*)) - \lambda_2 \sin^2(\text{dist}(x, x^*))$$

$$= \lambda_1 \sin^2 \text{dist}(x, x^*)) - \lambda_2 \sin^2(\text{dist}(x, x^*)) = \delta \sin^2(\text{dist}(x, x^*)).$$

Since $x \in B_a$ with $a > 0$, we have that $x$ and $x^*$ are in the same hemisphere and thus $d = \text{dist}(x, x^*) \leq \pi/2$. The desired result follows using $\sin(\phi) \geq \phi/2$ for $0 \leq \phi \leq \pi/2$. $\qquad\square$

**Proposition 3.** *A function $f: \mathbb{S}^{d-1} \to \mathbb{R}$ which satisfies quadratic growth with constant $\mu > 0$ and is 2a-weakly-quasi-convex with respect to a minimizer $x^*$ in a ball of this minimizer, satisfies for any $x$ in the same ball the inequality*

$$f(x) - f^* \leq \frac{1}{a}\langle \text{grad} f(x), -\log_x(x^*)\rangle - \frac{\mu}{2}\text{dist}^2(x, x^*).$$

*Proof.* From quadratic growth and weak-quasi-convexity, we have

$$\frac{\mu}{2}\text{dist}^2(x, x^*) \leq f(x) - f^* \leq \frac{1}{2a}\langle \text{grad} f(x), -\log_x(x^*)\rangle.$$

Now, again by weak-quasi-convexity

$$f(x) - f^* \leq \frac{1}{2a}\langle \text{grad} f(x), -\log_x(x^*)\rangle + \frac{\mu}{2}\text{dist}^2(x, x^*) - \frac{\mu}{2}\text{dist}^2(x, x^*)$$

$$\leq \frac{1}{a}\langle \text{grad} f(x), -\log_x(x^*)\rangle - \frac{\mu}{2}\text{dist}^2(x, x^*)$$

by substituting the previous inequality. $\qquad\square$

**Proposition 4.** *The function $f(x) = -x^T A x$ satisfies*

$$\|\operatorname{grad} f(x)\|^2 \geq \delta a^2 (f(x) - f^*)$$

*for any $x \in B_a$ with $a > 0$.*

*Proof.* By Proposition 3, we have

$$f(x) - f^* \leq \frac{1}{a} \langle \operatorname{grad} f(x), -\log_x(x^*) \rangle - \frac{\delta}{4} \operatorname{dist}^2(x, x^*)$$

since, in our case, $\mu = \delta/2$. Using $\langle x, y \rangle \leq \frac{1}{2}(\|x\|^2 + \|y\|^2)$ for all $x, y \in \mathbb{R}^d$, we can write for any positive $\rho$ that

$$\langle \operatorname{grad} f(x), -\log_x(x^*) \rangle \leq \frac{\rho}{2} \|\operatorname{grad} f(x)\|^2 + \frac{1}{2\rho} \|\log_x(x^*)\|^2.$$

Combining with $\rho = \frac{2}{a\delta}$ and using (6), we get

$$f(x) - f^* \leq \frac{1}{a} \frac{1}{a\delta} \|\operatorname{grad} f(x)\|^2 + \frac{1}{a} \frac{a\delta}{4} \operatorname{dist}^2(x, x^*) - \frac{\delta}{4} \operatorname{dist}^2(x, x^*) = \frac{1}{a^2\delta} \|\operatorname{grad} f(x)\|^2. \quad \square$$

**Proposition 5.** *The function $f(x) = -x^T A x$ is geodesically $2\lambda_1$-smooth on the sphere.*

*Proof.* The proof can be found also in [14, Lemma 1]. For $p \in \mathbb{S}^{d-1}$ and $v \in T_p\mathbb{S}^{d-1}$ with $\|v\| = 1$, we have that the Riemannian Hessian of $f$ satisfies

$$\langle v, \nabla^2 f(p)v \rangle = \langle v, -(I - pp^T)2Av + p^T 2Apv \rangle = -2v^T Av + 2p^T Ap \leq 2\lambda_1$$

because $v^T A v \geq 0$ (since $A$ is symmetric and positive semi-definite) and $p^T A p \leq \lambda_1$, by the definition of eigenvalues and $\|p\| = 1$. We have also used that $\|v\| = 1$ and $\langle p, v \rangle = 0$. Notice now that the largest eigenvalue of the Hessian is the maximum of $\langle v, \nabla^2 f(p)v \rangle$ over all $v \in T_p\mathbb{S}^r$ with $\|v\| = 1$, thus less or equal than $2\lambda_1$. This result easily implies Def. 12, since the Riemannian Hessian is the covariant derivative of the Riemannian gradient. $\quad \square$

**Proposition 6.** *An iterate of Riemannian gradient descent for $f(x) = -x^T A x$ starting from a point $x^{(t)} \in B_a$ and with step-size $\eta \leq a/\gamma$ where $\gamma \geq 2\lambda_1$, produces a point $x^{(t+1)}$ that satisfies*

$$\operatorname{dist}^2(x^{(t+1)}, x^*) \leq (1 - a\mu\eta) \operatorname{dist}^2(x^{(t)}, x^*) \qquad \text{for } \mu = \delta/2.$$

*Proof.* By definition of $x^{(t+1)}$, we have $\log_{x^{(t)}}(x^{(t+1)}) = -\eta \operatorname{grad} f(x^{(t)})$. Applying Proposition 16, we can thus write

$$\begin{aligned}
\operatorname{dist}^2(x^{(t+1)}, x^*) &\leq \| -\eta \operatorname{grad} f(x^{(t)}) - \log_{x^{(t)}}(x^*) \|^2 \\
&= \eta^2 \|\operatorname{grad} f(x^{(t)})\|^2 + \|\log_{x^{(t)}}(x^*)\|^2 + 2\eta \langle \operatorname{grad} f(x^{(t)}), \log_{x^{(t)}}(x^*) \rangle.
\end{aligned}$$

By Proposition 3 and 5, we have

$$\begin{aligned}
\frac{1}{a} \langle \operatorname{grad} f(x^{(t)}), \log_{x^{(t)}}(x^*) \rangle &\leq f^* - f(x^{(t)}) - \frac{\mu}{2} \operatorname{dist}^2(x^{(t)}, x^*) \\
&\leq -\frac{1}{2\gamma} \|\operatorname{grad} f(x^{(t)})\|^2 - \frac{\mu}{2} \operatorname{dist}^2(x^{(t)}, x^*).
\end{aligned}$$

Multiplying with $2\eta a$ and using $\eta \leq a/\gamma$, we get

$$\begin{aligned}
2\eta \langle \operatorname{grad} f(x^{(t)}), \log_{x^{(t)}}(x^*) \rangle &\leq -\frac{\eta a}{\gamma} \|\operatorname{grad} f(x^{(t)})\|^2 - \mu\eta a \operatorname{dist}^2(x^{(t)}, x^*) \\
&\leq -\eta^2 \|\operatorname{grad} f(x^{(t)})\|^2 - \mu\eta a \operatorname{dist}^2(x^{(t)}, x^*).
\end{aligned}$$

Substituting to the first inequality, we get the desired result. $\quad \square$

## C  Convergence of Distributed Gradient Descent on the Sphere

**Lemma 8.** *If $\eta \leq \frac{\cos(D)}{\gamma}$, the previous quantized gradient descent algorithm produces iterates $x^{(t)}$ and quantized gradients $q^{(t)}$ that satisfy*

$$\text{dist}^2(x^{(t)}, x^*) \leq \xi^t D^2, \quad \|q_i^{(t)} - \text{grad} f_i(x^{(t)})\| \leq \frac{\theta R^{(t)}}{2n}, \quad \|q^{(t)} - \text{grad} f(x^{(t)})\| \leq \theta R^{(t)}.$$

*Proof.* We do the proof by induction. We start from the case that $t = 0$. The first inequality is direct by the definition of $D$.

For the second one we have

$$\|\text{grad} f_i(x^{(0)}) - \text{grad} f_{i_0}(x^{(0)})\| \leq \|\text{grad} f_i(x^{(0)})\| + \|\text{grad} f_{i_0}(x^{(0)})\| \leq 2\gamma\pi$$

and by the definition of quantization, we get

$$\|\text{grad} f_i(x^{(0)}) - q_i^{(0)}\| \leq \frac{\theta R^{(0)}}{2n}.$$

Similarly for the third one, we have

$$\|\text{grad} f(x^{(0)}) - r^{(0)}\| \leq \sum_{i=1}^n \|\text{grad} f_i(x^{(0)}) - q_i^{(0)}\| \leq \frac{\theta R^{(0)}}{2}.$$

Then,

$$\|r^{(0)} - \text{grad} f_i(x^{(0)})\| \leq \|r^{(0)} - \text{grad} f(x^{(0)})\| + \|\text{grad} f(x^{(0)}) - \text{grad} f_i(x^{(0)})\| \leq \frac{\theta R^{(0)}}{2} + 2\pi\gamma.$$

By the definition of the quantization, we have

$$\|q^{(0)} - \text{grad} f(x^{(0)})\| \leq \frac{\theta R^{(0)}}{2} \leq \theta R^{(0)}.$$

We assume now that the inequalities hold for $t$ and we wish to prove that they continue to hold for $t + 1$.

We start with the first one and denote by $\tilde{x}^{(t+1)}$ the iteration of exact gradient descent starting from $x^{(t)}$. Since $\text{dist}(x^{(t)}, x^*) \leq D$, we have that $x^{(t)} \in B_a$ with $a = \cos(D)$.

We have

$$\text{dist}(x^{(t+1)}, x^*) \leq \text{dist}(x^{(t+1)}, \tilde{x}^{(t+1)}) + \text{dist}(\tilde{x}^{(t+1)}, x^*) \leq \|\eta \text{grad} f(x^{(t)}) - \eta q^{(t)}\| + \sqrt{\sigma}\text{dist}(x^{(t)}, x^*).$$

We have the last inequality, because $\text{dist}(\tilde{x}^{(t+1)}, x^*) \leq \sqrt{\sigma}\text{dist}(x^{(t)}, x^*)$ by Proposition 6 and $\text{dist}(x^{(t+1)}, \tilde{x}^{(t+1)}) \leq \|\log_{x^{(t)}}(x^{(t+1)}) - \log_{x^{(t)}}(\tilde{x}^{(t+1)})\| = \|\eta \text{grad} f(x^{(t)}) - \eta q^{(t)}\|$ by Proposition 16.

Thus

$$\text{dist}(x^{(t+1)}, x^*) \leq \frac{a}{\gamma}\theta R^{(t)} + \sqrt{\sigma}(\sqrt{\xi})^t D \leq \theta K(\sqrt{\xi})^t D + \sqrt{\sigma}(\sqrt{\xi})^t D \leq (\theta K + \sqrt{\sigma})(\sqrt{\xi})^t D \leq (\sqrt{\xi})^{t+1} D$$

which concludes the induction for the first inequality.

For the second inequality, we have

$$
\begin{aligned}
\|\text{grad} f_i(x^{(t+1)}) - \Gamma_{x^{(t)}}^{x^{(t+1)}} q_i^{(t)}\| &\leq \|\text{grad} f_i(x^{(t+1)}) - \Gamma_{x^{(t)}}^{x^{(t+1)}} \text{grad} f_i(x^{(t)})\| + \|\Gamma_{x^{(t)}}^{x^{(t+1)}} \text{grad} f_i(x^{(t)}) - \Gamma_{x^{(t)}}^{x^{(t+1)}} q_i^{(t)}\| \\
&\leq \gamma_i \text{dist}(x^{(t+1)}, x^{(t)}) + \|\text{grad} f_i(x^{(t)}) - q_i^{(t)}\| \\
&\leq 2\frac{\gamma}{n}(\sqrt{\xi})^t D + \theta\frac{R^{(t)}}{n} = 2\frac{\gamma}{n}(\sqrt{\xi})^t D + \theta\gamma K(\sqrt{\xi})^t D/n \\
&= (2/K + \theta)K\gamma(\sqrt{\xi})^t D/n \leq (\sqrt{\sigma} + \theta K)K\gamma(\sqrt{\xi})^t D/n = \frac{R^{(t+1)}}{n}
\end{aligned}
$$

and by the definition of the quantization scheme, we have

$$\|\mathrm{grad}\, f_i(x^{(t+1)}) - q_i^{(t+1)}\| \le \frac{\theta R^{(t+1)}}{2n}.$$

For the third inequality, we have

$$\|r^{(t+1)} - \mathrm{grad}\, f(x^{(t+1)})\| \le \sum_{i=1}^{n} \|q_i^{(t+1)} - \mathrm{grad}\, f_i(x^{(t+1)})\| \le \frac{\theta R^{(t+1)}}{2}$$

and

$$
\begin{aligned}
\|r^{(t+1)} - \Gamma^{x^{(t+1)}}_{x^{(t)}} q^{(t)}\| &\le \|r^{(t+1)} - \mathrm{grad}\, f(x^{(t+1)})\| + \|\mathrm{grad}\, f(x^{(t+1)}) - \Gamma^{x^{(t+1)}}_{x^{(t)}} \mathrm{grad}\, f(x^{(t)})\| \\
&\quad + \|\Gamma^{x^{(t+1)}}_{x^{(t)}} \mathrm{grad}\, f(x^{(t)}) - \Gamma^{x^{(t+1)}}_{x^{(t)}} q^{(t)}\| \\
&\le \frac{\theta R^{(t+1)}}{2} + \gamma \mathrm{dist}(x^{(t+1)}, x^{(t)}) + \theta R^{(t)} \\
&\le \frac{\theta R^{(t+1)}}{2} + R^{(t+1)} = \left(1 + \frac{\theta}{2}\right) R^{(t+1)}
\end{aligned}
$$

by using again the argument for deriving the second inequality. The last inequality implies that

$$\|\mathrm{grad}\, f(x^{(t+1)}) - q^{(t+1)}\| \le \frac{\theta R^{(t+1)}}{2}$$

by the definition of quantization. Summing the two last inequalities completes the induction. $\qquad \square$

**Theorem 9.** *Let $\eta \le \frac{\cos(D)}{\gamma}$. Then, the previous quantized gradient descent algorithm needs at most*

$$b = \mathcal{O}\left(nd \frac{1}{\cos(D)\delta\eta} \log\left(\frac{n}{\cos(D)\delta\eta}\right) \log\left(\frac{D}{\epsilon}\right)\right)$$

*bits in total to estimate the leading eigenvector with an accuracy $\epsilon$ measured in intrinsic distance.*

*Proof.* For computing the cost of quantization at each step, we use Proposition 7.

The communication cost of encoding each $\mathrm{grad}\, f_i$ at $t = 0$

$$\mathcal{O}\left(d \log \frac{2\gamma\pi}{\frac{\theta R^{(0)}}{2n}}\right) = \mathcal{O}\left(d \log \frac{4n\gamma\pi}{\theta\gamma K\pi}\right) \le \mathcal{O}\left(d \log \frac{2n}{\theta}\right).$$

Now we use that $\sigma \ge \frac{1}{2}$ and have

$$\frac{1}{\theta} = \frac{4}{\sqrt{\sigma}(1 - \sqrt{\sigma})} \le \frac{12}{1 - \sigma} = \frac{12}{\cos(D)\eta\mu}.$$

Thus, the previous cost becomes

$$\mathcal{O}\left(d \log \frac{2\gamma\pi}{\frac{\theta R^{(0)}}{2n}}\right) = \mathcal{O}\left(d \log \frac{n}{\cos(D)\eta\mu}\right).$$

The communication cost of deconding each $q_i^{(0)}$ in the master node is

$$\mathcal{O}\left(d \log \frac{2\gamma\pi + \frac{\theta R^{(0)}}{2}}{\frac{\theta R^{(0)}}{2}}\right) \le \mathcal{O}\left(d \log \frac{2\gamma\pi}{\frac{\theta R^{(0)}}{2}}\right) = \mathcal{O}\left(d \log \frac{1}{\cos(D)\eta\mu}\right).$$

This is because $2\gamma\pi \ge \frac{\theta R^{(0)}}{2}$.

Thus, the total communication cost at $t = 0$ is

$$\mathcal{O}\left(nd \log \frac{n}{\cos(D)\eta\mu}\right).$$

For $t > 0$, the cost of encoding $\operatorname{grad} f_i$'s is

$$\mathcal{O}\left(nd\log\frac{R^{(t+1)}/n}{\theta R^{(t+1)}/2n}\right) = \mathcal{O}\left(nd\log\frac{2}{\theta}\right) = \mathcal{O}\left(nd\log\frac{1}{\cos(D)\eta\mu}\right).$$

as before.

The cost of decoding in the master node is

$$\mathcal{O}\left(nd\log\frac{(1+\theta/2)R^{(t+1)}}{\theta R^{(t+1)}/2}\right) = \mathcal{O}\left(nd\log\frac{1}{\theta}\right) = \mathcal{O}\left(nd\log\frac{1}{\cos(D)\eta\mu}\right).$$

because $\theta/2 \le 1$.

Thus, the cost in each round of communication is in general bounded by

$$\mathcal{O}\left(nd\log\frac{(1+\theta/2)R^{(t+1)}}{\theta R^{(t+1)}/2}\right) = \mathcal{O}\left(nd\log\frac{n}{\cos(D)\eta\mu}\right).$$

Our algorithm reaches accuracy $a$ in function values if

$$\operatorname{dist}(x^{(t)}, x^*) \le \epsilon.$$

We can now write

$$\operatorname{dist}^2(x^{(t)}, x^*) \le \xi^t D^2 \le e^{-(1-\xi)t}D^2.$$

Thus, we need to run our algorithm for

$$\mathcal{O}\left(\frac{1}{1-\xi}\log\frac{D}{\epsilon}\right) \le \mathcal{O}\left(\frac{1}{\cos(D)\mu\eta}\log\frac{D}{\epsilon}\right)$$

many iterates to reach accuracy $a$.

The total communication cost for doing that is

$$\mathcal{O}\left(\frac{1}{\cos(D)\mu\eta}\log\frac{D}{\epsilon}nd\log\frac{n}{\cos(D)\eta\mu}\right) = \mathcal{O}\left(nd\frac{1}{\cos(D)\mu\eta}\log\frac{n}{\cos(D)\mu\eta}\log\frac{D}{\epsilon}\right)$$

Substituting

$$\mu = \frac{\delta}{2}$$

by Proposition 2, we get

$$\mathcal{O}\left(nd\frac{1}{\cos(D)\delta\eta}\log\frac{n}{\cos(D)\delta\eta}\log\frac{D}{\epsilon}\right)$$

many bits in total. $\qquad\square$

## D    Initialization

### D.1    Uniformly Random Initialization

We estimate numerically the constant $c$ that is used in (3) of Section 5.1.

Let $x^{(0)}$ be chosen from a uniform distribution on the sphere $\mathbb{S}^{d-1}$. We are interested in $\alpha_1 = v_1^T x^{(0)}$ for some fixed $v_1 \in \mathbb{S}^{d-1}$. By spherical symmetry, $\alpha_1$ is distributed in the same way as the first component of $x^{(0)}$. Let $A_d(h)$ be the surface of the hyperspherical cap of $\mathbb{S}^{d-1}$ with height $h \in [0, 1]$. Then it is obvious that

$$\mathbb{P}(|\alpha_1| \ge a) = A_d(1-a)/A_d(1) = I_{1-a^2}(\tfrac{d-1}{2}, \tfrac{1}{2}),$$

where we used the well-known formula for $A_d(h)$ in terms of the regularized incomplete Beta function $I_x(a, b)$; see, e.g., [21]. Solving the above expression[2] for $a$ when it equals a given probability $1 - p$,

---

[2]This can be conveniently done using `https://docs.scipy.org/doc/scipy/reference/generated/scipy.special.betaincinv.html`

we can calculate the interval $[-1, -a] \cup [a, 1]$ in which $\alpha_1$ will lie for a random $x^{(0)}$ up to probability $1 - p$.

In the figure below, we have plotted these values of $a$ divided by $p/\sqrt{d}$ for $p = 10^{-1}, 10^{-2}, 10^{-3}, 10^{-4}$. Numerically, there is strong evidence that $a \geq cp/\sqrt{d}$ with $c = 1.25$.

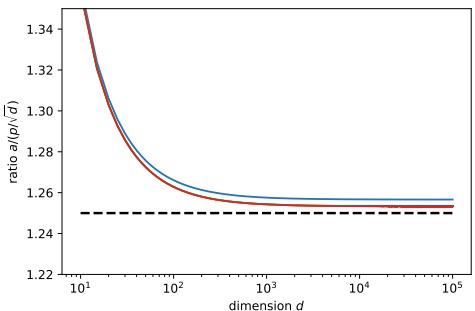

## D.2 Initialization in the Leading Eigenvector of the Master

**Proposition 10.** *Assume that our data are i.i.d. and sampled from a distribution $\mathcal{D}$ bounded in $\ell_2$ norm by a constant $h$. Given that the eigengap $\delta$, the number of machines $n$ and the total number of data points $m$ satisfy*

$$\delta \geq \Omega\left(\sqrt{m}\sqrt{n}\sqrt{\log \frac{d}{p}}\right), \tag{4}$$

*we have that the previous quantization costs $\mathcal{O}(nd)$ many bits and $\langle x^{(0)}, x^* \rangle$ is lower bounded by a constant with probability at least $1 - p$.*

*Proof.* By Lemma 3 in [14] we have that

$$\left\| A_{i_0} - \frac{1}{m}\sum_{i=1}^{n} A_i \right\|^2 \leq \frac{32 \log\left(\frac{d}{p}\right) h^2}{m_{i_0}}$$

which implies that

$$\left\| m A_{i_0} - \sum_{i=1}^{n} A_i \right\|^2 \leq 32 \frac{m^2}{m_{i_0}} \log\left(\frac{d}{p}\right) h^2 = 32mn \log\left(\frac{d}{p}\right) h^2$$

with probability at least $1 - p$. Of course $\sum_{i=1}^{n} A_i = A$.

From this bound we can derive a bound for the distance between the eigenvectors of the two matrices. Indeed, using Lemmas 5 and 8 in [14], we can derive

$$1 - \langle v^{i_0}, x^* \rangle \leq \frac{\sqrt{128mn \log\left(\frac{d}{p}\right)} h}{\delta}$$

and

$$\langle v^{i_0}, x^* \rangle \geq 1 - \frac{\sqrt{128mn \log\left(\frac{d}{p}\right)} h}{\delta}$$

with probability at least $1 - p$ (note that the leading eigenvector of $A_{i_0}$ is equal to the leading eigenvector $n A_{i_0}$). This is because $\langle v^{i_0}, x^* \rangle \leq 1$, which implies that $\langle v^{i_0}, x^* \rangle^2 \leq \langle v^{i_0}, x^* \rangle$.

We notice that the squared distance of $v^{i_0}$ and $x^*$ is

$$\|v^{i_0} - x^*\|^2 = \|v^{i_0}\|^2 + \|x^*\|^2 - 2\langle v^{i_0}, x^* \rangle = 2(1 - \langle v^{i_0}, x^* \rangle) \leq 2\frac{\sqrt{128mn \log\left(\frac{d}{p}\right)} h}{\delta}$$

which is upper bounded by a constant by Assumption 4. The same holds for $\|v^i - x^*\|$, thus, by triangle inequality we have an upper bound on $\|v^{i_0} - v^i\|$ to be at most double of the upper bound for $\|v^{i_0} - x^*\|$, thus it is still upper bounded by a constant. Since $\langle v^{i_0}, x^* \rangle$ is lower bounded by a constant, again by Assumption 4, we have that the ratio of the input to the output variance in the quantization of $v^{i_0}$ is upper bounded by a constant. Thus, the total communication cost of this quantization is $\mathcal{O}(nd)$.

By the definition of the quantization scheme, we get

$$\|\tilde{x}^{(0)} - v^{i_0}\| \leq \frac{\langle v^{i_0}, x^* \rangle}{2(\sqrt{2} + 2)} =: \zeta.$$

For the projected vector $x^{(0)}$, we have

$$\|x^{(0)} - v^{i_0}\| \leq \|\tilde{x}^{(0)} - v^{i_0}\| + \|\tilde{x}^{(0)} - x^{(0)}\| \leq 2\|\tilde{x}^{(0)} - v^{i_0}\| \leq 2\zeta$$

because $x^{(0)}$ is the closest point to $\tilde{x}^{(0)}$ belonging to the sphere and $v^{i_0}$ belongs also to the sphere.

By the triangle inequality, we have

$$\|x^{(0)} - x^*\| \leq \|v^{i_0} - x^*\| + \|x^{(0)} - v^{i_0}\|$$

which is equivalent to

$$\sqrt{2(1 - \langle x^{(0)}, x^* \rangle)} \leq \sqrt{2(1 - \langle v^{i_0}, x^* \rangle)} + 2\zeta.$$

Thus

$$\langle x^{(0)}, x^* \rangle \geq \langle v^{i_0}, x^* \rangle - \sqrt{2(1 - \langle v^{i_0}, x^* \rangle)}\zeta - 2\zeta^2 \geq \langle v^{i_0}, x^* \rangle - \sqrt{2}\zeta - 2\zeta$$

$$= \langle v^{i_0}, x^* \rangle - (\sqrt{2} + 2)\zeta = \langle v^{i_0}, x^* \rangle - (\sqrt{2} + 2)\frac{\langle v^{i_0}, x^* \rangle}{2(\sqrt{2} + 2)} = \frac{\langle v^{i_0}, x^* \rangle}{2}$$

with probability at least $1 - p$.

Since $\langle v^{i_0}, x^* \rangle$ is lower bounded by a constant, $\langle x^{(0)}, x^* \rangle$ is also lower bounded by a constant and we get the desired result.

$\square$