# OpenReview forum: "Distributed Principal Component Analysis with Limited Communication"
_NeurIPS.cc/2021/Conference — NeurIPS 2021 Poster_

### Official Review · Reviewer_F3cp · 2021-07-16

**Rating:** 6
**Confidence:** 3

**Summary:**

This paper studies the problem of finding the leading eigenvector in a distributed setting, where the $i$-th worker holds $m_i$ pieces of local data and the goal is to find the leading eigenvector of the global covariance data matrix while minimizing the total communication cost. Unlike most of the previous results that largely focus on round complexity, this paper considers bit complexity.

The proposed algorithm is a combination of the Riemannian gradient descent over an $\ell_2$ sphere and the vector quantization method based on Davies et al. (2021). The main technical contribution of this paper is showing a "strong-weak-convexity" result of the quadratic function $-x^TAx$ over a Riemannian manifold, which resolves a technical issue in Huang et al. (2020).

**Limitations And Societal Impact:**

The authors have listed the limitations as part of future works. This work does not present any foreseeable societal consequences due to its theoretical nature.

**Main Review:**

Though this paper fixes a technical issue in the previous work Huang et al. (2020),  I do feel the novelty is rather limited since both the Riemannian gradient descent and the vector quantization technique are well-studied before. In addition, I have the following concerns/suggestions:

1. The comparison with prior works can be improved. For instance, although previous works mainly focus on the round complexity, one can apply standard vector quantization techniques to convert the round complexity to the bit complexity. The authors should compare their proposed scheme with this baseline.


2. Following the previous point, to study the round complexity of the proposed scheme, one can check in how many iterations the Riemannian terminates. However, in the Algorithm section (Page 6) the authors did not specify the stopping criteria.


3. The final bit complexity involves a term $\gamma$ (which is defined as $n\max \lambda_i$). To better compared with other results, the authors can show how it scales as a function of $n, b$ under the i.i.d. setting (and perhaps assume $m_i = m$).


4. An experimental result to verify the proposed scheme would highly strengthen the paper.


5. The presentation can be improved. For instance, in the main body, several important definitions that appear in the appendix are referenced. Some of them should be moved to the main paper. Also, the algorithm section on Page 6 can be made more concise.


6. A minor typo in line 77: $A = \sum_{i}^n M_i^TM_i$ instead of  $\sum_{i}^d M_i^TM_i$.




**Time Spent Reviewing:**

6

---

> ### Author Response · Authors · 2021-08-10
> **Answer to F3cp**
>
> Novelty is limited: We do not agree that the novelty of the work is limited. Firstly, to the best of our knowledge, we present the first Riemannian distributed algorithm with limited communication. In order to be able to add quantization to gradient descent, we need a contraction-style convergence rate like the one of Proposition 6. All the other gradient-descent-like methods for leading eigenvector computation have a convergence rate similar to the power method, which is not guaranteed to be contracting. In order to guarantee contraction (Proposition 6), we rely on weak-quasi-convexity and combine it with a quadratic growth property, strengthening the results of [30]. Since contraction at each step is a desirable property in optimization in general, this result is of general interest besides distributed PCA with limited communication. Our paper can trivially also provide a round complexity for the same problem (we have one round of communication at each iteration). Of course there are points for future work, such as improving the step-size of GD while approaching the minimizer and also extending the framework to full PCA (Grassmann/Stiefel manifold), and we currently work on them.
>
> Round Complexity to Bit Complexity: This could be a reasonable baseline for experiments, however we note that there is no guarantee that quantizing in such a naive way would yield an algorithm with convergence guarantees.
>
> Stopping Criteria: We run the algorithm up to accuracy $\epsilon$ (which of course is not known in advance). We can also count round complexity with the same criterion. This is standard practice in theoretical analysis of optimization methods in general but also in quantized methods (one says that the iteration complexity of GD for reaching accuracy $\epsilon$ in the minimization of a convex function is $O(1/\epsilon)$). In the practical implementation of the algorithms one can either run for a fixed number of rounds (in which case the theoretical analysis provides a bound on the final accuracy), or terminate after a threshold on the norm of the Riemannian gradient (in our case quantized approximation of the gradient). The latter could considerably reduce the number of iterations compared to our theoretical bound but is not easy to analyse. Our experiments show that (at least on Gaussian data) convergence is generally smooth and predictable, and therefore any reasonable stopping criterion should be effective.
>
> Term $\gamma$ in complexity bound:
> The term $\gamma$ (smoothness constant) is optimally equal to $2\lambda_1$ and $2 n \max {\lambda_i} = n \max {\gamma_i}$ is a naive upper bound for theoretical purposes. Usually we have much better estimation for $\gamma$ in practice. However it is probably possible to derive a better upper bound for $\gamma$ based on the assumptions of Proposition 10 and we will investigate that for the next version (thanks for the suggestion).
>
> Definitions from the Appendix:
> We tried to refer consistently to the appendix for any knowledge of differential geometry needed. However, you are right that this breaks the fluency of the text and, since space permits, we will try to incorporate important definitions in the main text.
>
> The paper fixes a technical issue in the previous work [Huang et al. (2020)]:
> We believe this statement is somewhat inexact, as the technical details of our approach are rather different relative to Huang et al. (2020), and we are as of yet unaware of a way to fix the technical issue we pointed out in this reference. We just mention that a robust algorithm like gradient descent (with a proper step-size), together with weak-convexity and quadratic growth can guarantee contraction towards the optimum at each iterate, thus a valid theoretical analysis in contrast with [Huang et al. (2020)].

---

> > ### Comment · Reviewer_F3cp · 2021-08-31
> > **To the authors**
> >
> > Thanks for adding the experimental results and clarifying the relation with [Huang et al. (2020)]. My main concern is still the comparison with previous works.
> >
> > In particular, in terms of the bit-complexity, i agree with the authors that directly quantizing the DSI method [Garber et al. (2017)] may not guarantees convergence (though it may still be worth to compare to DSI with full precision both theoretically and empirically), but the authors can also add some comparisons in terms of the round-complexity. According to the proof of Theorem 9, the algorithm stops in $\tilde{O}\left( \frac{\delta}{a\eta}\right) \approx \tilde{O}\left( \frac{\gamma}{a^2\delta}\right) $ (picking $\eta = \frac{a}{\gamma}$ as suggested by Thm 9) rounds. Since in the worst-case $\gamma = O(n\max \lambda_i)$, the resulting round-complexity can be $\tilde{O}\left( \frac{n\max \lambda_i}{a^2\delta}\right) $, which seems to be way larger than the guarantees in [Garber et al. (2017)].
> >
> > I understand that this paper mainly focuses on bit-complexity instead of round-complexity, but if the proposed algorithm has good bit-complexity, it should also give a reasonable round-complexity (instead of off by a factor of $n$). I would appreciate if the authors can comment on this point.

---

> > > ### Author Response · Authors · 2021-09-01
> > > **good question**
> > >
> > > We thank the reviewer for the interesting question. To address it, we note the following three distinct points:
> > >
> > > 1. First, we emphasize that Garber et al. work under stronger assumptions, which allow them to get lower round complexity compared to standard methods: they assume that all machines draw data i.i.d. from _the same_ distribution. Although sometimes assumed in distributed optimization, this is a non-trivial limitation; to our reading, without this assumption, their method would probably not work.
> > >
> > > 2. Second, we discuss the round complexity of our method in detail, emphasizing inherent round complexity trade-offs due to quantization.
> > >    Indeed, the round complexity of our quantized RGD algorithm until reaching accuracy $\epsilon$ is $\tilde O( \frac{\gamma}{a^2 \delta})$. Specifically, if the initialization $x^{(0)}$ is uniformly random in the sphere (Section 5.1), then $a=O(\frac{1}{\sqrt{d}})$ . If we have warm start (Section 5.2), then $a=O(1)$. (Both statements hold with high probability.)
> > >
> > >    Firstly, we would like to point out that theoretically $\gamma=2 \lambda_1$ (see Proposition 5) and that this parameter should be used for comparison with related work. The bound $\gamma \leq n \max{\gamma_i}$ is just a  (worst case) distributed estimation of the same quantity with limited communication. In most previous works, the value of $\lambda_1$ is assumed to be known in advance. This should be a reasonable assumption, since practically we can have very good estimations, but we also wanted to provide a theoretically rigorous upper bound.
> > >    Thus, our round complexity is $\tilde O(d\frac{\lambda_1}{\delta})$ with random initialization and can be improved to $\tilde O(\frac{\lambda_1}{\delta})$ with warm start and other assumptions (e.g. i.i.d. data, as in Garber et al.)
> > >
> > >    For random initialization, the round complexity of the distributed power method is $\tilde O(\frac{\lambda_1}{\delta})$. (Please see the table at the beginning of page 4 in Garber et al.). This is because the convergence rate of the power method is
> > >    $
> > >      \tan(\angle (x^{(t)},x^*)) \leq (1-\lambda_1/\delta)^t \tan(\angle (x^{(0)},x^*))
> > >    $
> > >    which is not distance-contracting because $\tan(\angle(p,q))$ is not a distance function between $p$ and $q$. The dimension is again present in that rate, because $ \tan(\angle (x^{(0)},x^*))= \Theta(\sqrt{d})$ (with high probability), but this appears inside a logarithm in the final iteration complexity.
> > >
> > >    Since we want to insert quantization, the standard--and as far as we know, only--framework to accomplish this is by starting from a method that has a *contracting* convergence rate (see Magnusson et al.) in terms of intrinsic distances $\text{dist}(x^{(t)},x^*)=\angle(x^{(t)},x^*)$ (see our Proposition 6). It may be possible to add quantization in a non-contracting algorithm, but we are unaware of such a technique. Thus, we firstly prove that RGD with step-size $\frac{a}{\gamma}$ is contracting (based on weak-convexity and quadratic growth) and then we add quantization, paying the price of having an extra $O(1/a^2)=O(d)$ in our round complexity. Thus, our round complexity is slightly worse than the distributed power method, but this is in some sense inherent due to the introduction of quantization. Despite this, our algorithm can achieve close to optimal bit complexity.
> > >
> > >
> > >
> > > 3. Third, upon careful thought, we realized that in fact we can reduce this extra $O(d)$ term in our round complexity to $O(\sqrt{d})$, by carefully adjusting the step-size. The intuition is that we take $a \rightarrow 1$, while $x^{(t)} \rightarrow x^*$.
> > >    We present a detailed sketch of this round-efficient extension of our algorithm in the (anonymized) write-up below:
> > >
> > >    https://www.dropbox.com/s/pq0axvl5k9j7nms/pca-round-efficient-sketch.pdf
> > >
> > >    This argument also improves our bit complexity in Section 5.1 by an $\sqrt{d}$ factor ($d^2$ becomes $d^{3/2}$).

---

> > > > ### Comment · Reviewer_F3cp · 2021-09-01
> > > > **Increase my score**
> > > >
> > > > Thanks for the detailed response, it addressed my concerns and i will increase my score accordingly. I would suggest the authors to include the above comparisons with previous works (i.e. in terms of the round-complexity) in the next revision.

---

> > > > > ### Author Response · Authors · 2021-09-01
> > > > > **thanks**
> > > > >
> > > > > Thank you for increasing your score, this is indeed a point that must be discussed and we will add a small section about it in our next version.

---

### Official Review · Reviewer_m9Bn · 2021-07-16

**Rating:** 6
**Confidence:** 3

**Summary:**

The paper considers the problem of distributed computation of the leading eigenvector of a data covariance matrix. To solve this problem, a distributed variant of Riemannian gradient descent is proposed. Convergence is proven and an upper bound is given on the number of bits that need to be transmitted to estimate the leading eigenvector. Finally, the dependency of the method on its initialization is analyzed.


**Limitations And Societal Impact:**

The authors discuss limitations of the work in the last section of the paper. Since this is a methodological contribution, there is no direct societal impact.

**Main Review:**

Originality:

The paper presents a novel algorithm for the computation of the leading eigenvector of a covariance matrix, as it is done in principal component analysis. The paper fills a gap in the literature since compared to previous approaches, the proposed technique is specifically designed to minimize the bandwidth cost, i.e. the number of bits transmitted between the nodes. The main contribution of the paper is the theoretical analysis of the approach, in particular the convergence is proven and a bound on its communication complexity is given.

Quality:

The derivation of the algorithm is technically sound, and the authors show theoretically that the approach is indeed communication efficient. However, there is no experimental evaluation of the approach. For a paper proposing a novel algorithm, I would expect at least some experiments that show its usefulness in practice. Moreover, while the approach is motivated by the application to principal component analysis, the paper only focuses on computing the first eigenvector, while in practice typically multiple eigenvectors are required.

Clarity:

The paper is clearly written and well-organized. However, as stated above, a section containing the experimental validation of the approach as well as details on the implementation are missing.

Significance:

Due to the ever increasing amounts of data, coming up with novel distributed algorithms is a worthy goal. As such, the paper makes an important contribution to the field, in particular since it is specifically designed to be efficient in terms of communication complexity. Moreover, the distributed variant of Riemannian gradient descent may turn out to be useful in different contexts. On the other hand, the paper appears to be incomplete as it lacks any sort of experimental evaluation. Moreover, for the approach to be adopted for PCA it should be extended to compute multiple eigenvectors of the covariance matrix.


Post Rebuttal:

My main concern was the lack of experimental evaluation, which was addressed by the authors in their rebuttal. I have therefore increased my score.

**Time Spent Reviewing:**

5

---

> ### Author Response · Authors · 2021-08-10
> **Answer to m9Bn**
>
> Practical Implementation: As mentioned, we have implemented our proposal, and compare experimentally with Riemannian Gradient Descent and variants. The report is available here:
>
> https://pdfhost.io/v/eCrEPV9AE_experimentsA.pdf
>
> Limitation of computing only first leading eigenvector: Firstly, we would like to note that leading eigenvector computation is itself an important problem, and many papers in the literature deal with that. See for instance [13] and [Efficient coordinate-wise leading eigenvector computation, Wang et al.]. The connection to full PCA can be done in two ways:
> - Compute other eigenvectors one-by-one while deflating the previously calculated eigenvectors (subtract from the covariance matrix the rank-one terms in the spectral decomposition). This should work directly with the theory presented in the current paper by minor adjustment of the cost function and Lipschitz constants. However, it requires the assumption that every eigenvalue is sufficiently separated since otherwise the eigenvalue gaps used in the complexity bound will be very small. This is a very unrealistic assumption for interior eigenvalues in real-world data. We therefore did not include such an analysis in the paper.
> - Compute the eigenvectors as a subspace, either on the Stiefel or Grassmann manifold. Judging from classical eigenvalue algorithms (subspace iteration), it should only require a gap with the last eigenvalue. This can be done with Riemannian optimization but requires a lot of new analysis and belongs to our plans for future work.

---

> ### Author Response · Authors · 2021-09-04
> **Final follow-up**
>
> Dear Reviewer,
>
> As the discussion period is drawing to a close, we would like to gently ask if it would be possible to examine our rebuttal with respect to the experimental results and discussion of extensions full PCA, and determine whether it could allow a reconsideration of your verdict on our submission.

---

### Official Review · Reviewer_YLwt · 2021-07-23

**Rating:** 6
**Confidence:** 4

**Summary:**

This paper proposed a communication efficient distributed algorithm to compute the leading eigenvector of a covariance data matrix where the rows of the matrix are distributed on multiple nodes. They presented a distributed Riemannian gradient descent to compute the leading eigenvector and avoid projection onto constraint operation. To prove the convergence of Riemannian gradient descent, they first derive the weakly-quasi-convex property of the objective.  A quantization scheme is also proposed to save communication costs. Finally, they proved the convergence of the algorithm, and the communication complexities for different initialization schemes.


**Limitations And Societal Impact:**

1. For the quantization algorithm, it will be great if the authors can give the explicit mapping rule in the main paper, since the readers may not be familiar with lattice quantization.

2. I would like the authors to discuss the relation between this work and federated PCA [Grammenos, Andreas, et al.19], where they also consider data matrix is distributed on different nodes.


3. I would like to see some experiments on the effectiveness of this algorithm, so that we can see how the quantization idea improves the communication cost compared to the non-quantized version.


**Main Review:**

Distributed eigenvector estimation is important in many distributed ML algorithm's implementation, and communication cost is also a pain point in distributed learning. This paper, for the first time, tackles these two aspects together. Using Riemannian gradient descent to optimize manifold-constrained problems is a popular idea, and so is the quantized idea to save communication cost. The authors are the first to combine these ideas and apply them to distributed PCA. The structure of this paper is clear and easy to follow. I believe a reader without prior knowledge can follow this paper. The paper is well-organized and the presentation of the theorem/lemma is clear.


**Time Spent Reviewing:**

2

---

> ### Author Response · Authors · 2021-08-10
> **Answer to YLwt**
>
> Explicit Quantization Mapping: This is not easy to do in general, since we need to create a new lattice in the tangent space based at each point x_t. [Davies et al.] provides several options for this, with differing properties and computational costs.
> In practice, we use a random cubic lattice in the relevant tangent space, which means that all lattice operations involved reduce to simple and inexpensive coordinate-wise (in the lattice basis) rounding procedures, while only losing small (logarithmic) factors in the theoretical guarantees, as shown in [Davies et al.]. We will add a paragraph explaining this procedure along with the experiments section in the updated version.
>
> Federated PCA: We were not aware of this paper, thanks for pointing it out. We will include it in our related work section.
>
> Experiments vs the non-quantized algorithm: As mentioned in our general comment, we compare experimentally with full-precision Riemannian Gradient Descent and reach similar performance using only 2-4 bits per coordinate, i.e., an 8x to 16x  communication reduction. The report is available here:
>
> https://pdfhost.io/v/eCrEPV9AE_experimentsA.pdf

---

### Author Response · Authors · 2021-08-10
**General Answer to Reviews**

We would like to thank the reviewers for their time reviewing this work. The reviewers seem to appreciate the theoretical contribution of the paper, while the most significant concern is the lack of experimental verification of our main algorithm. Firstly, we would like to point out that our main motivation is theoretical, and more precisely to develop the first algorithm for distributed leading eigenvector computation with limited communication.

However, to address the reviewers’ main concern, we have implemented the algorithm, and have compared against Riemannian Gradient Descent, using both un-quantized full-precision gradients, and naive Euclidean gradient quantization, as suggested by the reviewers. You can find the report in the link: https://pdfhost.io/v/eCrEPV9AE_experimentsA.pdf. These initial experiments are on synthetic Gaussian data, but we plan to extend them to real data sets for the updated version and make our code publicly available. The results show that our approach of quantizing Riemannian gradients in the tangent space performs significantly better than standard Euclidean quantization, and approaches the performance of full-precision gradient descent using only 2-4 bits per coordinate, i.e., an 8x to 16x  communication reduction.

---

### Author Response · Authors · 2021-08-20
**Discussion Follow-up**

Dear Reviewers and ACs,

Since more than a week has passed since our response, we would like to follow-up on it. We believe our rebuttal does address the issues raised in a substantive manner (e.g. in the form of experimental results validating our approach), and we would be very glad to have the opportunity to further clarify these or any other issues that the reviewers may have.

Best regards,
The Authors

---

### Decision · Program_Chairs · 2021-09-27

**Decision:**

Accept (Poster)

**Comment:**

All reviewers support acceptance for the contributions on communication-efficient distributed PCA and the theoretical aspects of the proposed approach. I also recommend acceptance. However, please make sure to incorporate experimental results in the final version to validate the theoretical findings. A discussion on how to compute multiple eigenvectors would also strengthen the paper.